# Self-reported disability in relation to mortality in rural Malawi: a longitudinal study of over 16 000 adults

Josephine E Prynn ,[1,2] Albert Dube,[2] Joseph Mkandawire,[2] Olivier Koole,[3] Steffen Geis,[2,3] Elenaus Mwaiyeghele,[2] Oddie Mwiba,[2] Alison J Price,[3] Lackson Kachiwanda,[2] Moffat Nyirenda,[3,4] Hannah Kuper ,[5] Amelia C Crampin[2,3]

[1]Institute of Cardiovascular Science, University College London, London, UK
[2]Malawi Epidemiology and Intervention Research Unit, Lilongwe, Malawi
[3]Faculty of Infectious and Tropical Diseases, London School of Hygiene and Tropical Medicine, London, UK
[4]NCD Phenotype Programme, MRC/UVRI Uganda Research Unit on AIDS, Entebbe, Wakiso, Uganda
[5]Faculty of Epidemiology and Population Health, London School of Hygiene and Tropical Medicine, London, UK

**Correspondence to**
Dr Josephine E Prynn;
josephineprynn@gmail.com

## ABSTRACT

**Objectives** We investigated whether self-reported disability was associated with mortality in adults in rural Malawi.

**Setting** Karonga Health and Demographic Surveillance Site (HDSS), Northern Malawi.

**Participants** All adults aged 18 and over residing in the HDSS were eligible to participate. During annual censuses in 2014 and 2015, participants were asked if they experienced difficulty in any of six functional domains and were classified as having disabilities if they reported 'a lot of difficulty' or 'can't do at all' in any domain. Mortality data were collected until 31 December 2017. 16 748 participants (10 153 women and 6595 men) were followed up for a median of 29 months.

**Primary and secondary outcome measures** We used Poisson regression to examine the relationship between disability and all-cause mortality adjusting for confounders. We assessed whether this relationship altered in the context of obesity, hypertension, diabetes or HIV. We also evaluated whether mortality from non-communicable diseases (NCD) was higher among people who had reported disability, as determined by verbal autopsy.

**Results** At baseline, 7.6% reported a disability and the overall adult mortality rate was 9.1/1000 person-years. Adults reporting disability had an all-cause mortality rate 2.70 times higher than those without, and mortality rate from NCDs 2.33 times higher than those without.

**Conclusions** Self-reported disability predicts mortality at all adult ages in rural Malawi. Interventions to improve access to healthcare and other services are needed.

## Strengths and limitations of this study

► We were able to use mortality data collected through a well-functioning health and demographic surveillance survey in a country without complete vital registration.

► The large sample size allowed us to examine the relationship between disability and mortality over a relatively short follow-up period.

► This is the first study to examine whether disability as defined by the Washington Group questions is predictive of mortality, so contributes substantially to the existing literature on this topic.

► A limitation of the study is that we have considerable missing data in chronic health states (body mass index, hypertension, diabetes, HIV), which limits our understanding of how these are related to disability and mortality, and that verbal autopsies were used to establish cause of death.

education, employment and social life,[1] and are at high risk of poverty.[2] Consequently, disability is recognised as an important development issue, and is explicitly referenced within five of the Sustainable Development Goals (SDG).[3]

Disability is expected to be linked to increased mortality through various pathways. Both disability and mortality are related to ageing and poverty, and so associations may arise through confounding.[1 2] An underlying disease may also cause both disability and mortality, for instance, diabetes can lead to impairments (eg, visual impairment from diabetic retinopathy) and premature death. People with disabilities may find it more difficult to seek healthcare due to a range of barriers,[1] thereby increasing their mortality risk. They may also have unhealthier behaviours, on average, regarding smoking, physical activity and diet, increasing their vulnerability to obesity, non-communicable diseases (NCD) and mortality.[4] There is also evidence that people with disabilities are

## INTRODUCTION

Disability is a complex concept encompassing long-term physical, mental, intellectual or sensory impairments which, in interaction with personal and environmental factors, may limit people's participation in society on an equal basis with others.[1] The WHO estimates that 1 billion people globally have a disability,[1] and the number is expected to rise further as the global population continues growing, life expectancy rises and average age increases. People with disabilities are being 'left behind' in terms of inclusion in

more vulnerable to HIV and other infectious diseases, due to their marginalised position in society.[5] Certain impairments, such as mobility or cognitive impairments, can increase frailty, depression and functional difficulties, all linked to increased mortality.[6–8] Data are, however, lacking on the link between disability and mortality, particularly from low and middle-income settings, where over 80% of people with disabilities reside.[9] One reason for the lack of data has been due to lack of assessment of disability in surveys and cohorts. The debate has raged as to how disability should be defined and measured, yet scales are needed to estimate the prevalence and impacts of disability, and to allow international data comparisons. Consensus is growing on the use of the Washington Group (WG) Short Set to collect Disability Statistics,[10] which will improve data comparability. The WG questions focus on difficulties in functioning in six different domains related to activities (eg, walking) and participation (eg, performing usual activities).

The objective of this study was to investigate the relationship between self-reported disability and mortality among adults in rural Malawi. A secondary objective was to assess whether the effect of disability on mortality varies with coexistence of obesity or a chronic disease (hypertension, diabetes or HIV), and by cause of death.

## METHODS
### Setting and data collection

The Karonga Health and Demographic Surveillance Site (HDSS) in Northern Malawi comprises a population of around 40 000 individuals, which is largely representative of the rural Malawian population in terms of age and sex structure.[11 12] Census information is collected on the population annually, along with continuous reporting of births, deaths and migration by community key informants. Verbal autopsy is done after every death using a semistructured interview of a family member using an adaptation of a WHO instrument.[13] Two clinically trained reviewers independently assign cause of death based on this interview. In case of disagreement, a third reviewer arbitrates.

Since 2014, the WG Short Set questions on disability have been added to the annual census questionnaire, alongside the existing questions on demographic, health and social indicators. The disability questions were only asked if the participant was physically seen by the fieldworker, although they could be answered through a proxy, after obtaining written consent or assent. Therefore, no disability data were collected from anyone away from home on the day of the census.

The questions, translated into the local language of Chitumbuka, are:

▶ Do you have difficulty seeing, even if wearing glasses?
▶ Do you have difficulty hearing, even if using a hearing aid?
▶ Do you have difficulty walking or climbing steps?
▶ Do you have difficulty remembering or concentrating?
▶ Do you have difficulty (with self-care such as) washing all over or dressing?
▶ Using your usual (customary) language, do you have difficulty communicating, for example, understanding or being understood?

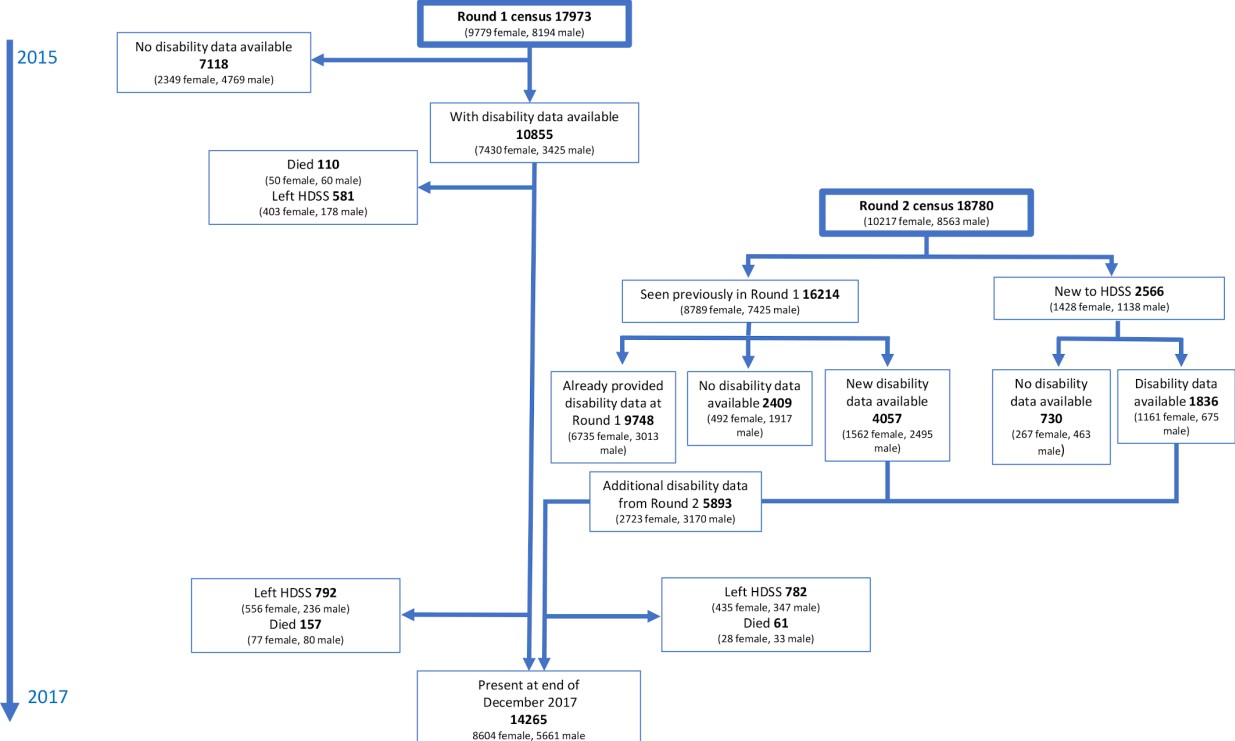

**Figure 1** Flow chart of participation. HDSS, Karonga Health and Demographic Surveillance Site.

In this analysis, we used data on adults aged 18 and over from the first two consecutive census rounds to include the disability questions. The first was in 2014–2015 (round 1), the second in 2015–2016 (round 2). Baseline disability status was taken from the round 1 census data where possible. For participants with no disability data from round 1, data from round 2 were used. Anyone who moved into the HDSS between rounds 1 and 2, or turned 18 between rounds 1 and 2, and answered disability questions at round 2 was also included. Other sociodemographic information was taken from the same interview as the disability data. Follow-up was undertaken until 31 December 2017.

Data on body mass index (BMI), hypertension and diabetes were gathered from a survey of all adults performed within the HDSS population in 2013–2015 on prevalence of major NCDs and their risk factors.[14] In this screening, height and weight were measured twice, and the mean was used to calculate BMI. Participants were asked if they had previously been diagnosed with hypertension or diabetes, and whether they were taking any regular medication. Resting blood pressure was measured three times with 5 min in between, and a mean of the second and third measures was used. Blood was taken for plasma glucose measurement after at least 8 hours of fasting. These data were collected a mean of 1.2 years prior to the disability data (maximum 3.5 years).

Data on HIV status were collated from numerous sources including an HIV serosurvey in 2011 and the 2013–2015 NCD survey. Data on new HIV diagnoses were also collected from consenting participants at government clinics within the HDSS. Participants were categorised as HIV positive if they had ever reported a diagnosis of HIV or had a positive antibody test. They were categorised as negative for 4 years after a negative HIV test, after which time their status was categorised as missing in case of a new infection in the interim.

### Variables

Age was grouped into categories of 18–34, 35–44, 45–54, 55–64, 65–69, 70–74, 75–79 and 80+ years; narrower age bands were chosen at higher ages as self-reported disability is strongly associated with older age. Education level was defined as no education, primary (including partially completed), secondary (including partially completed) and tertiary. Occupation was categorised into not working (including unemployed, unable to work and retired), manual work, farming or fishing and non-manual work (including professional work). Participants were defined as in a union if they were married or cohabiting, and not if single, divorced or widowed.

Categories of BMI were <18.5 kg/m² (underweight), 18.5–24.9 kg/m² (healthy weight), 25–29.9 kg/m² (overweight) and 30+ kg/m² (obese). Hypertension was defined as systolic blood pressure ≥140 mm Hg, diastolic blood pressure of ≥90 mm Hg or use of antihypertensive medications. Diabetes was defined as fasting blood sugar ≥7.0 mmol/L or a self-reported diagnosis of diabetes.

Disability was defined as answering 'a lot of difficulty' or 'can't do at all' in any domain.

Cause of death among adults was broadly categorised into communicable disease, NCDs, maternal death, external (including injury and poisoning) and unspecifiable/other.

### Statistical analysis

Prevalence of self-reported disability was calculated with 95% CIs by sociodemographic and health characteristics. Poisson regression analysis was used to calculate adult mortality rate ratios (RR), comparing people reporting 'some difficulty' and 'a lot of difficulty or can't do at all' with 'no difficulty' in each disability domain. For these analyses, individuals contributed exposure time during their residence in the HDSS from the date the disability survey was completed, until the earliest of 31 December 2017, death or outmigration. Returning and repeat migrants only contributed person-years while resident in the HDSS.

Age and sex were included a priori in the adjusted model, and baseline occupation, education level and union status were sequentially added to the model to check for confounding. Any variable that altered the RR more than 10% was kept in the adjusted model. Each of obesity, hypertension, diabetes and HIV status at baseline was also added to the models to check for confounding or effect modification. Cause-specific mortality was also calculated. We performed complete case analysis, so that any participants with missing data for any of the variables in the model were excluded. Sensitivity analyses, including an 'unknown' category for BMI, hypertension, diabetes and HIV were also performed.

All significance tests were likelihood ratio tests.

### Patient and public involvement

Malawi Epidemiology and Intervention Research Unit works closely with the community in which this research was conducted. Regular meetings with senior community members take place to ensure that study objectives align with the priorities of the community, and that the methodology and procedures are appropriate and acceptable. Research findings are disseminated similarly.

### RESULTS

Of 17 973 adult (≥18 years) residents seen at round 1, a total of 10 855 (60.4%) responded to questions on disability. A further 5893 adults not included at round 1 (because they were under 18 years, not living in the area at round 1 or missed) provided disability data at round 2, giving 16 748 participants with baseline disability data (figure 1).

Those without disability data were more likely to be younger (22.3% of 18–39 year-olds compared with 10.8% of those aged 40+) and male (29.4% of men compared with 9.5% of women). Baseline characteristics of the census population compared with those included in the

**Table 1** Baseline characteristics of participants with baseline disability data and proportion with self-reported disability†

| | Women | | | Men | | | Total | | |
|---|---|---|---|---|---|---|---|---|---|
| | n | Reporting disability, n (%) | P value* | n | Reporting disability, n (%) | P value* | n | Reporting disability, n (%) | P value* |
| Overall | 10 153 | 857 (8.4) | – | 6595 | 420 (6.4) | – | 16 748 | 1277 (7.6) | – |
| Age group | | | | | | | | | |
| 18–34 | 5503 | 167 (3.0) | <0.001* | 3378 | 73 (2.2) | <0.001* | 8881 | 240 (2.7) | <0.001* |
| 35–44 | 1914 | 103 (5.4) | | 1339 | 48 (3.6) | | 3253 | 151 (4.6) | |
| 45–54 | 1100 | 122 (11.1) | | 786 | 55 (7.0) | | 1886 | 177 (9.4) | |
| 55–64 | 773 | 116 (15.0) | | 486 | 64 (13.2) | | 1259 | 180 (14.3) | |
| 65–69 | 258 | 54 (20.9) | | 175 | 25 (14.3) | | 433 | 79 (18.2) | |
| 70–74 | 212 | 75 (35.4) | | 133 | 38 (28.6) | | 345 | 113 (32.8) | |
| 75–79 | 189 | 94 (49.7) | | 138 | 45 (32.6) | | 327 | 139 (42.5) | |
| 80+ | 204 | 126 (61.8) | | 160 | 72 (45.0) | | 364 | 198 (54.4) | |
| Education level‡ | | | | | | | | | |
| None | 378 | 117 (31.0) | 0.002* | 98 | 27 (27.6) | 0.06* | 476 | 144 (30.3) | <0.001* |
| Some primary/ completed primary | 6603 | 632 (9.6) | | 3284 | 269 (8.2) | | 9887 | 901 (9.1) | |
| Some secondary/ completed secondary | 2633 | 82 (3.1) | | 2707 | 99 (3.7) | | 5340 | 181 (3.4) | |
| Tertiary | 469 | 15 (3.2) | | 467 | 24 (5.1) | | 936 | 39 (4.2) | |
| Occupation§ | | | | | | | | | |
| Not working | 1013 | 202 (19.9) | <0.001 | 985 | 106 (10.8) | <0.001 | 1998 | 308 (15.4) | <0.001 |
| Manual | 143 | 5 (3.5) | | 745 | 28 (3.8) | | 888 | 33 (3.7) | |
| Farmer/fisherman | 7637 | 567 (7.4) | | 4092 | 256 (6.3) | | 11 729 | 823 (7.0) | |
| Non-manual/ business/ professional | 1249 | 62 (5.0) | | 724 | 26 (3.6) | | 1973 | 88 (4.5) | |
| Union status¶ | | | | | | | | | |
| Not in a union | 3381 | 479 (14.2) | <0.001 | 1933 | 97 (5.0) | 0.001 | 5314 | 576 (10.8) | <0.001 |
| In a union | 6768 | 378 (5.6) | | 4659 | 323 (6.9) | | 11 427 | 701 (6.1) | |
| BMI (kg/m²)** | | | | | | | | | |
| <18.5 | 669 | 61 (9.1) | 0.005 | 601 | 57 (9.5) | 0.38 | 1270 | 118 (9.3) | <0.001 |
| 18.5–24.9 | 6074 | 423 (7.0) | | 4847 | 263 (5.4) | | 10 921 | 686 (16.3) | |
| 25–29.9 | 1869 | 176 (9.4) | | 497 | 50 (10.1) | | 2366 | 226 (9.6) | |
| 30+ | 689 | 90 (13.1) | | 80 | 10 (12.5) | | 769 | 100 (13.0) | |
| Hypertension†† | | | | | | | | | |
| No hypertension | 6069 | 416 (6.9) | 0.69 | 3599 | 200 (5.6) | 0.05 | 9668 | 616 (6.4) | 0.05 |
| Hypertension | 1011 | 242 (23.9) | | 633 | 110 (17.4) | | 1644 | 352 (21.4) | |
| Diabetes‡‡ | | | | | | | | | |
| No diabetes | 6189 | 535 (8.6) | 0.22 | 3582 | 259 (7.2) | 0.007 | 9771 | 794 (8.1) | 0.008 |
| Diabetes | 109 | 25 (22.9) | | 69 | 19 (27.5) | | 178 | 44 (24.7) | |
| HIV status§§ | | | | | | | | | |
| Negative | 6128 | 490 (8.0) | 0.69 | 3070 | 218 (7.1) | 0.60 | 9198 | 708 (7.7) | 0.90 |
| Positive | 907 | 70 (7.7) | | 443 | 38 (8.6) | | 1350 | 108 (8.0) | |
| Chronic diseases (n)¶¶ *** | | | | | | | | | |

Continued

**Table 1** Continued

| | Women | | | Men | | | Total | | |
|---|---|---|---|---|---|---|---|---|---|
| | n | Reporting disability, n (%) | P value* | n | Reporting disability, n (%) | P value* | n | Reporting disability, n (%) | P value* |
| 0 | 4243 | 261 (6.2) | 0.77* | 2054 | 102 (5.0) | 0.002* | 6297 | 363 (5.8) | 0.02* |
| 1 | 1166 | 170 (14.6) | | 620 | 91 (14.7) | | 1786 | 261 (14.6) | |
| 2 | 100 | 19 (19.0) | | 69 | 14 (20.3) | | 169 | 33 (19.5) | |
| 3 | 5 | 1 (20.0) | | 2 | 1 (50.0) | | 7 | 2 (28.6) | |

*P value for difference using likelihood ratio test, values marked as * (age, education level, number of chronic diseases) p value for trend. For all variables other than age group this is controlled for age (as a continuous variable).
†Defined as answering 'a lot of difficulty' or 'can't do at all' to at least one disability domain.
‡Education level: no data on 70 women and 39 men.
§Occupation: no data on 111 women and 49 men.
¶Union status: no data on 4 women and 3 men.
**BMI: no data on 872 women and 570 men.
††Hypertension: no data on 3073 women and 2363 men.
‡‡Diabetes: no data on 3855 women and 2944 men.
§§HIV status: no data on 3118 women and 3082 men.
¶¶Chronic diseases included in this variable: hypertension, diabetes, HIV.
***Number of chronic diseases variable: no data on 4639 women and 3850 men (only participants with data on all three chronic diseases included).
BMI, body mass index.

study at Round 1 are found in online supplementary table S1. Participants with disability data had a median follow-up of 29 months (IQR 20–33 months). Migration out of the HDSS led to censoring of data in 1574 (9.5%) of participants. Individuals who migrated were more likely to be young and have higher educational attainment.

Table 1 shows that most participants were under age 35, and 76.1% of women and 62.5% of men were farmers or fishermen. The prevalence of obesity was higher in women than men (7.4% vs 1.3%), whereas prevalence of hypertension, diabetes and HIV was similar among men and women. Hypertension, diabetes and HIV variables were not available for at least 30% of participants.

Prevalence of self-reported disability in one or more domains was 7.6% and strongly associated with increasing age (p test for trend <0.001). Adjusted for age, people with lower levels of education and those not working were more likely to report disability, and prevalence was higher among participants with obesity, hypertension or diabetes. People living with HIV reported a similar prevalence of disability to those who were HIV negative.

There were 328 deaths in 36 019 person-years of follow-up. The adult mortality rate was 9.1/1000 person-years (95% CI 8.2 to 10.1) and higher among men (12.9/1000 person-years; 95% CI 11.1 to 15.0) than women (6.8/1000 person-years; 95% CI 5.8 to 8.0) including when controlling for age (RR 1.75; 95% CI 1.44 to 2.14). Sensitivity analyses comparing all-cause mortality rates in men without disability data to those with disability data suggested no material difference stratified by age (online supplementary table S2).

Table 2 shows that compared with adults reporting 'no difficulty' in any disability domain, those reporting 'some difficulty' and 'a lot of difficulty or can't do at all' had greater mortality (RR 1.39; 95% CI 1.03 to 1.88 and RR 2.70; 95% CI 1.91 to 3.82, respectively). Except for difficulty seeing and hearing, self-reported disability in any of the domains was strongly associated with increased mortality when adjusted for age, and the association remained when also adjusted for sex and occupation. For every domain, reporting disability ('a lot of difficulty' or 'can't do at all') was associated with a higher mortality than answering 'some difficulty'. Disability in increasing numbers of domains was strongly associated with mortality. The relationship between any self-reported disability and all-cause mortality did not vary by sex or age group (table 3); and none of BMI, hypertension, diabetes or HIV were confounders or effect modifiers of the relationship (online supplementary table S3). Sensitivity analysis including a 'missing' category for hypertension did show heterogeneity by hypertension status (p=0.03): the relationship between disability and mortality was stronger among those with unknown hypertension status (online supplementary table S4). Including a 'missing' category for BMI, diabetes and HIV did not show any heterogeneity.

Table 4 shows that mortality from NCDs was higher among people living with disability than those without disability. No difference was seen for mortality from HIV or other communicable diseases.

## DISCUSSION

Our findings show that self-reported disability in adults is strongly associated with increased all-cause mortality in rural Malawi. Furthermore, a stepwise increase in risk was

**Table 2** Poisson regression analysis of effect of graded levels of self-reported disability on all-cause adult mortality

| | Deaths (n) | Person-years | Model 1: adjusted for age only* | | Model 2: adjusted for age, sex and occupation† | |
|---|---|---|---|---|---|---|
| | | | RR (95% CI) | P value‡ | RR (95% CI) | P value‡ |
| **Difficulty seeing** | | | | | | |
| No difficulty | 165 | 28106 | 1 | 0.38 | | 0.85 |
| Some difficulty | 107 | 6316 | 1.02 (0.79 to 1.34) | | 1.08 (0.82 to 1.41) | |
| A lot of difficulty/can't do at all | 49 | 1255 | 1.20 (0.84 to 1.73) | | 1.01 (0.7 to 1.46) | |
| **Difficulty hearing** | | | | | | |
| No difficulty | 296 | 33922 | 1 | 0.05 | | 0.19 |
| Some difficulty | 39 | 1480 | 1.20 (0.85 to 1.71) | | 1.11 (0.78 to 1.58) | |
| A lot of difficulty/can't do at all | 13 | 275 | 1.79 (1.01 to 3.15) | | 1.51 (0.85 to 2.68) | |
| **Difficulty walking** | | | | | | |
| No difficulty | 157 | 30222 | 1 | <0.001 | | <0.001 |
| Some difficulty | 79 | 4175 | 1.39 (1.04 to 1.88) | | 1.57 (1.15 to 2.13) | |
| A lot of difficulty/can't do at all | 85 | 1278 | 3.40 (2.48 to 4.66) | | 3.20 (2.27 to 4.51) | |
| **Difficulty remembering** | | | | | | |
| No difficulty | 204 | 29608 | 1 | <0.001 | 1 | <0.001 |
| Some difficulty | 79 | 5384 | 1.06 (0.81 to 1.39) | | 1.17 (0.89 to 1.54) | |
| A lot of difficulty/can't do at all | 37 | 650 | 2.67 (1.84 to 3.85) | | 2.55 (1.74 to 3.73) | |
| **Difficulty communicating** | | | | | | |
| No difficulty | 287 | 35235 | 1 | <0.001 | 1 | <0.001 |
| Some difficulty | 23 | 332 | 3.11 (2.01 to 4.8) | | 2.26 (1.44 to 3.53) | |
| A lot of difficulty/can't do at all | 10 | 82 | 8.81 (4.67 to 16.59) | | 5.33 (2.78 to 10.21) | |
| **Difficulty with self-care** | | | | | | |
| No difficulty | 234 | 33850 | 1 | <0.001 | 1 | <0.001 |
| Some difficulty | 47 | 1306 | 1.84 (1.33 to 2.56) | | 1.74 (1.24 to 2.45) | |
| A lot of difficulty/can't do at all | 38 | 503 | 3.75 (2.62 to 5.38) | | 3.14 (2.15 to 4.6) | |
| **Any disability** | | | | | | |
| No difficulty | 84 | 22197 | 1 | <0.001 | 1 | <0.001 |
| Some difficulty | 122 | 10645 | 1.39 (1.03 to 1.88) | | 1.61 (1.18 to 2.19) | |
| A lot of difficulty/can't do at all | 115 | 2839 | 2.70 (1.91 to 3.82) | | 2.65 (1.84 to 3.8) | |
| **Disability in multiple domains** | | | | | | |
| No disability | 206 | 32848 | 1 | <0.001 | 1 | <0.001 |
| Disability in one domain | 51 | 2018 | 1.60 (1.15 to 2.21) | | 1.51 (1.08 to 2.11) | |
| Disability in two domains | 31 | 543 | 2.39 (1.59 to 3.59) | | 2.07 (1.35 to 3.17) | |
| Disability in three domains | 19 | 120 | 3.71 (2.26 to 6.09) | | 3.06 (1.83 to 5.14) | |
| Disability in 4+ domains | 14 | 79 | 5.64 (3.19 to 9.96) | | 4.33 (2.39 to 7.88) | |

*Controlled for age (continuous variable).
†Controlled for age (continuous variable), sex and occupation (not working, manual work, farming/fishing, non-manual work).
‡Likelihood ratio test for difference.

observed with increasing levels of difficulties in functioning and numbers of domains affected. Individuals reporting difficulty in walking and self-care were particularly at risk. The coexistence of obesity or chronic disease did not materially alter estimates of effect, and the magnitude of the association was consistent across age and sex groups. People dying from NCDs were more likely to have been living with disability before death than those dying from HIV.

Our findings are similar to those observed elsewhere in sub-Saharan Africa (SSA). A study in Tanzanian adults aged 70+ also found a stepwise increase in mortality with increasing disability severity.[15] In South African adults

**Table 3** Poisson regression of disability* on all-cause adult mortality stratified by age group and sex†

| Age group | | Women | | | | | Men | | | | |
|---|---|---|---|---|---|---|---|---|---|---|---|
| | | Deaths (n) | PY | Mortality rate per 1000 PY (95% CI) | RR‡ (95% CI) | P value | Deaths (n) | PY | Mortality rate per 1000 PY (95% CI) | RR‡ (95% CI) | P value |
| 18–49 | No disability | 40 | 16741 | 2.39 (1.75 to 3.26) | 1 | 0.04 | 51 | 9870 | 5.17 (3.93 to 6.80) | 1 | 0.10 |
| | Disability | 6 | 739 | 8.12 (3.65 to 18.09) | 2.76 (1.16 to 6.58) | | 5 | 324 | 15.43 (6.42 to 37.06) | 2.37 (0.94 to 5.97) | |
| 50–69 | No disability | 25 | 3169 | 7.89 (5.33 to 11.67) | 1 | 0.002 | 44 | 1952 | 22.54 (16.78 to 30.29) | 1 | 0.06 |
| | Disability | 14 | 566 | 24.74 (14.65 to 41.77) | 3.00 (1.55 to 5.79) | | 12 | 255 | 46.98 (26.68 to 82.73) | 1.93 (1.02 to 3.65) | |
| 70+ | No disability | 18 | 749 | 24.04 (15.15 to 38.15) | 1 | <0.001 | 29 | 636 | 45.62 (31.70 to 65.64) | 1 | 0.009 |
| | Disability | 52 | 673 | 77.30 (58.90 to 101.45) | 2.76 (1.61 to 4.75) | | 32 | 337 | 95.07 (67.23 to 134.44) | 1.98 (1.19 to 3.30) | |
| All | No disability | 83 | 20659 | 4.02 (3.24 to 4.98) | 1 | <0.001 | 124 | 12458 | 9.95 (8.35 to 11.87) | 1 | 0.001 |
| | Disability | 72 | 1977 | 36.42 (28.91 to 45.88) | 2.89 (1.20 to 4.19) | | 49 | 916 | 53.49 (40.42 to 70.77) | 1.87 (1.30 to 2.71) | |

*Disability is defined as reporting 'a lot of difficulty' or 'can't do at all' in any of the functional domains.
†P value for heterogeneity between age and sex groups=0.94.
‡Rate ratio, adjusted for age (as a continuous variable).
PY, person-years.

**Table 4** Cause-specific mortality rate ratio among those living with disability prior to death compared with those not living with disability

| Cause of death* | Died (n) | | Mortality rate ratio† | P value |
|---|---|---|---|---|
| | **Disability** | **No disability** | | |
| Communicable disease (excluding HIV) | 17 | 12 | 1.42 (0.62–3.26) | 0.41 |
| HIV | 52 | 10 | 1.80 (0.89–3.65) | 0.10 |
| Non-communicable disease | 80 | 64 | 2.33 (1.60–3.38) | <0.01 |
| Maternal | 1 | 0 | – | – |
| External | 19 | 2 | 1.22 (0.32–4.65) | 0.77 |
| Unspecified/other | 7 | 10 | 3.92 (1.30–11.81) | 0.02 |

*Cause of death attributed by verbal autopsy.
†Adjusted for age (as a continuous variable).

aged 50+, severe functional disability was associated with a mortality RR of 1.92 in women and 2.80 in men, compared with those with least functional disability.[16] In Malawi, Payne et al described an increasing mortality risk with increasingly severe disability among adults aged 45 and over, compared with those without disability.[17] None of these studies used the WG questions, however.

The association between disability and mortality has also been observed in high-income countries (HIC).[18–21] This is seen even after adjusting for a wide range of comorbidities,[22–24] suggesting that the association between disability and mortality is not solely due to conditions predisposing to both disability and death. Nonetheless, contexts differ widely between high-resource settings and rural Malawi, hence these findings may not be generalisable to our setting. In HIC, improved access to healthcare means that many more people live with treated chronic disease over many years, as opposed to in a low-income country (LIC) like Malawi, where people may be more likely to die early from disabling disease.[14 25] Even access to safe water and sanitation services can be profoundly affected by disability in LICs including Malawi,[26 27] which is an important risk factor for disease.

We observed a higher prevalence of disability in women than men at all ages, yet mortality rates were consistently higher in men than women across all adult age groups, irrespective of disability. This apparent 'male-female health survival paradox' has been seen in a wide variety of settings, using a range of disability definitions.[15 16 23]

The population in Malawi is ageing, as in much of SSA.[28] As such, the very strong positive correlation between increasing age and prevalence of self-reported disability portends a growing prevalence of disability in the population in coming decades. The extent to which increasing life expectancy leads to an increased number of years lived with disability, or whether onset of disability will be delayed with a healthier life course leading to increased life expectancy, is not yet understood.[29]

A major strength of our study is the available mortality data from a long-established HDSS, when vital registration is very limited in Malawi.[30] Moreover, the large sample size and low loss to follow-up allows meaningful mortality analyses despite a relatively short follow-up time. Disability is not a static state: people move considerably between disability states over time.[17 31] Despite this, a one-off self-report of disability put individuals at a much-increased risk of death in the following 2 years. As further disability and mortality data are collected with time, we will be able to examine mortality over a longer follow-up time and consider self-reported disability as a time-varying exposure in future analyses. Our study adds considerably to the very limited literature examining mortality with disability in this context and is the first to examine whether disability as defined by the WG questions is predictive of mortality.

Our study has several limitations. The WG questions do not capture a comprehensive picture of disability, for example, common impairments such as affective disorders and chronic pain are not included. Furthermore, the use of self-report relies on participants having comparable interpretations of what constitutes 'some difficulty' versus 'a lot of difficulty'. Self-report also requires people to be able to understand the questions and communicate their response, so there is a risk that some participants would have been unable to self-identify. However, by allowing proxy responses where necessary, this will have been partially mitigated. We have limited information on underlying health conditions in the population and our data on hypertension, diabetes, BMI and HIV were not measured concurrently with disability. If participants were to change category in the intervening time misclassification could occur, but as these are all, in the main, long-term health states, it is unlikely that this will have affected a substantial number of participants. We had considerable missing data on hypertension, diabetes and HIV status, and our data may have been more likely to capture positive cases than negative. Moreover, it is likely that data were not missing at random, increasing the chance of bias in results. For hypertension and diabetes, the data were missing mainly on younger, working men, who are likely to have lower prevalence than the general population.[30] For HIV status some data are from new diagnoses made at health centres, where people may present if they feel they are at risk of being HIV positive. Participants with

missing data on hypertension demonstrated a stronger relationship between disability and mortality than those with known hypertension or known normotension, suggesting that the hypertension data are not missing at random. These potential biases limit our understanding of the relationship of these individual conditions with disability and mortality, as well as the composite measure of 'number of chronic conditions', and more research is needed to see whether our findings are replicated in both this and other settings. Furthermore, cause of death was collected through verbal autopsy, which is subject to bias. Our pragmatic approach to only collect disability reports on individuals found at home during our annual census means that older men were over-represented in our sample compared with the census. Consequently, the increased mortality in men compared with women seen in all age groups could be suspected to be due to a 'healthy worker' effect as more healthy men who will likely have lower mortality risk work far from the household in paid employment, while this pattern is not seen in women. However, we found that the relationship of disability and mortality did not vary by age group and that there were no differences in age-specific mortality by disability status when stratified by age. Particularly among younger men, the mortality rate was very similar among those included in this analysis and those excluded.

## CONCLUSIONS

Disability is commonly reported and likely to increase in Malawi in coming decades as the population ages and NCDs become increasingly prevalent. Self-reported disability is associated with significantly increased mortality, which seems to be driven particularly by an increased mortality rate from NCDs. While more research is urgently needed to help understand the mechanism behind this relationship, this needs to be concurrent with active interventions to improve access to healthcare and other services, particularly for people with disabilities, if the commitment to 'leave no-one behind'[32] in the SDGs is to be realised.

**Contributors** Conceptualisation: OK, AP, HK, AC. Funding acquisition: MN, AC. Design: AD, SG, AP. Project administration: AD, JM, EM, OM. Supervision of data collection: AD, OK, SG, EM, AP, LK. Data curation: OM. Analysis: JEP, AC. Original draft preparation: JEP, HK, AC. Review and editing of manuscript: JEP, AD, JM, OK, SG, EM, OM, AP, LK, MN, HK, AC.

**Funding** This work was supported by the Wellcome Trust (grant number 098610).

**Competing interests** None declared.

**Patient and public involvement** Patients and/or the public were involved in the design, or conduct, or reporting, or dissemination plans of this research. Refer to the Methods section for further details.

**Patient consent for publication** Not required.

**Ethics approval** Ethical approval for the HDSS census rounds and NCD survey was granted by the National Health Sciences Research Committee (NHSRC) (protocol numbers 419 and 1072, respectively), and by the London School of Hygiene and Tropical Medicine (LSHTM) (protocol numbers 5081 and 6303, respectively).

**Provenance and peer review** Not commissioned; externally peer reviewed.

**Data availability statement** Data are available upon reasonable request. Data from the NCD survey are available from LSHTM Data Compass: Malawi Epidemiology and Intervention Research Unit Non-Communicable Disease Survey

data, 2013–2017, https://doi.org/10.17037/DATA.0000096136. Data are available under the terms of the Creative Commons Attribution 3.0 International License (CC-BY 3.0). Summary demographic data sets are publicly available through the INDEPTH iShare platform. Longitudinal data (demographic surveillance episodes and linked rounds of disability questionnaires) cannot be sufficiently deidentified for public availability. Application may be made for access through the MEIRU director (mia.crampin@lshtm.ac.uk) or data scientist Chifundo Kanjala (chifundo.kanjala@lshtm.ac.uk). Those wishing to access the data will need to provide a brief proposal for what the data will be used for as a condition of access.

**ORCID iDs**
Josephine E Prynn http://orcid.org/0000-0002-5275-8644
Hannah Kuper http://orcid.org/0000-0002-8952-0023

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
