## [Reviewer comments · BMJ Open]

ARTICLE DETAILS

TITLE (PROVISIONAL)	Self-reported disability in relation to mortality in rural Malawi: a longitudinal study of over 16000 adults
AUTHORS	Prynn, Josephine; Dube, Albert; Mkandawire, Joseph; Koole, Olivier; Geis, Steffen; Mwaiyeghele, Elenaus; Mwiba, Oddie; Price, Alison; Kachiwanda, Lackson; Nyirenda, Moffat; Kuper, Hannah; Crampin, Amelia

VERSION 1 – REVIEW

REVIEWER	Emily Lauer Center for Developmental Disabilities Evaluation and Research, E. K. Shriver Center, University of Massachusetts Medical School USA
REVIEW RETURNED	11-Nov-2019

GENERAL COMMENTS	This paper presents analyses on an important topic regarding mortality patterns for adults with disabilities in a low income country. With revision, it will represent an important contribution to the literature. Many of the methods used in this paper are sound including the application of Washington Group questions, use of Poisson models for mortality analyses, use of sensitivity analyses, etc. However, there are some weaknesses with the manuscript described below which need to be addressed before it is of sufficient quality for publication. It's important that the writers clarify that the mortality rates they have calculated are adult mortality rates, not overall mortality rates. This is essential for clarity and comparison to other studies in the literature because the population is restricted to age 18 years and above which changes expectations for the mortality rate. The limitation section should include a discussion of the limits of self-report, including that people away from the home on the day of the census visit are not included. For example, some people with disabilities may not be able to respond, including because they may not be able to comprehend the questions, on a census survey. Therefore they may not be able to self-identify. The rate of missing disability data is also of concern because this data may not be missing at random in relation to disability status. I am also concerned about the difference in the sizes of the populations collected between genders - and am concerned that the healthy worker effect from the census methodology may be substantial. Are there other sources of data available to permit estimates of the proportion of males that may be missing?
---

	The considerable missing data is rightly noted as a limitation, but may not be emphasized strongly enough or with sufficient detail about its potential effects on the analytic results. The rate of missing data for some areas is substantial and has a strong likelihood of affecting the results. I am glad to see that sensitivity analyses were performed and reported with the covariates. The observed relationship with hypertension and missing data underscores that this missing data may not be at random across the subgroups, and may bias the results. This limitation section may need to be stated more strongly given these observations. Given the significant difference observed in the crude adult mortality rates between genders, it would be beneficial to age adjust one of the mortality rates to the other gender's age distribution to understand whether different age distributions within each gender may explain the observed difference. I note this type of sensitivity analysis was performed for men with and without disability, but not between genders. Given your later discussion of the survival paradox, it's to assess a potential role for different age distributions. For Table 4, the way the data is presented has the possibility to be misleading. Analyses of cause-specific mortality are an important area of analysis for this manuscript, especially as authors attempt to tie the paper's conclusions to it; the methodology for this table requires revision. The populations of people with no self-reported disability and self-reported disability are not of equal size, therefore, we would expect equal distribution within a cause of death category across these two groups. Because of this, simply comparing the percentages as done here is not informative. Instead, cause-specific adult mortality rates should be used for this comparison because this method will control for different underlying group sizes. A Risk Ratio or similar measure can be used to assess differences in cause-specific mortality rates between groups. The analyses in the paper don't directly support the conclusions. Disability does predict increased risk of mortality. However, the paper does not analyze why this association is present and does not directly identify healthcare access as a causal factor. I recognize that there are existing public health and access to healthcare challenges in this region, however, the analyses do not sufficiently link these challenges to the mortality outcomes, especially given the issues with the methodology used in Table 4. After revision of Table 4, authors should consider where there are differences in cause-specific mortality and improve the connection of the conclusions to these results. There are areas throughout the paper where the language could be more specific and clear. For example, in lines 18-23 on page 10 where sentences begin with "This association..." and "This suggests...". It would be more clear to limit the use of "this" and specify the subject of the sentence. This manuscript repeatedly states that the data is of high quality; I think is an overstatement of its quality. I understand that relative to other data sets from LIC countries it may be of higher quality. However, reliance on verbal autopsy and self-identification of disability does pose substantial risks to data quality. For example, certain causes of death are less likely to be identified through
--	---

	verbal autopsy, which likely biases this data. Additionally, missing information on 40% of the population is not trivial and as discussed in the limitation section, this missing data is likely not at random (due to difference in people not at home, etc.).
--	---

REVIEWER	Professor Richard Walker Northumbria Healthcare NHS Foundation Trust, UK
REVIEW RETURNED	19-Dec-2019

GENERAL COMMENTS	Self-reported disability in relation to mortality in rural Malawi: A longitudinal study of over 16000 adults Comments to authors This is an interesting study from the Karonga HDSS in northern Malawi in which, in the annual censuses in 2014 and 2015, adults aged 18 and over were asked whether they experienced difficulty in any of the 6 functional domains of the Washington Group (WG) short set to collect disability statistics. Mortality data were then collected up until the 31 December 2017. The 16748 participants were followed up for a median of 29 months. 7.6% of the population surveyed reported disability and the overall mortality rate was 9.4/1000 person-years. People reporting disability had mortality rate 2.7 times higher than those without and 44.4% of people who died from NCDs had disability compared to 16.1% of those who died from HIV. While this is a large and impressive study, and these data are unique, there are some limitations, most of which the authors acknowledge. Methods The WG short set questions on disability have been asked in each census since 2014. Have the authors compared the answers from individuals in one year with the next year to check for consistency in responses and whether these don't change over time for those with a "fixed" disability, and do in others, who may have a progressive problem? As the authors acknowledge excluding anyone who is away from home on the day of the census would mean that they were a lot more likely to miss men, particularly younger men, who may well have been away working. The questions in the WG short set questions could be interpreted in a subjective way with inter-individual differences in interpretation of what constitutes "some difficulty" versus "a lot of difficulty" for example as the authors acknowledge. Was this area covered within training workshops for the field staff relating to how they would interpret the questions? Also, as the authors acknowledge the WG questions do not capture a comprehensive picture of disability, as compared to longer tools. People who were not picked up in round 1 (2014/2015) could be included if they were picked up in round 2 (2015/16) if they had turned 18 between the time of the 2 censuses or had moved into the area. Those recruited in round 2 would not have been followed up for as long. The data on BMI, hypertension and diabetes were gathered in a survey between 2013 and 2015 and so are not contemporaneous,
---

	as acknowledged by the authors, with the disability survey. Also, as the authors acknowledge there are a lot of missing data for hypertension and diabetes as well as HIV status. This is a major limitation. In Table 1 no data on chronic disease were included for 4639 women and 3850 men as only participants with data on all three chronic diseases were included. Results I think that Figure 1 could be made a bit clearer. For the 39.6% who didn't respond to questions on disability in round 1 were their demographic characteristics different from the 60.4% who did respond? For the 5893 not included at round 1 who provided disability data at round 2, what percentage is this of the total population who could have been recruited at round 2? The fact that people could be recruited over 1 year apart would have greatly decreased the follow up time available for those recruited in round 2. The 21.9% migration out of the HDSS is very high for a relatively short time period.
--	---

VERSION 1 – AUTHOR RESPONSE

Dear De Lauer,

Many thanks for your considered comments on this submission. We have taken them into consideration and have updated the paper accordingly. In particular, we have updated the limitations section of the discussion and revisited Table 4 to make the analysis clearer.

Please find specific responses to each of your points below.

This paper presents analyses on an important topic regarding mortality patterns for adults with disabilities in a low income country. With revision, it will represent an important contribution to the literature. Many of the methods used in this paper are sound including the application of Washington Group questions, use of Poisson models for mortality analyses, use of sensitivity analyses, etc. However, there are some weaknesses with the manuscript described below which need to be addressed before it is of sufficient quality for publication.

It's important that the writers clarify that the mortality rates they have calculated are adult mortality rates, not overall mortality rates. This is essential for clarity and comparison to other studies in the literature because the population is restricted to age 18 years and above which changes expectations for the mortality rate. Thank you for this observation. We have now clarified this in the abstract, methods, and results sections.

The limitation section should include a discussion of the limits of self-report, including that people away from the home on the day of the census visit are not included. For example, some people with disabilities may not be able to respond, including because they may not be able to comprehend the questions, on a census survey. Therefore they may not be able to self-identify. We agree this is important to include and we have now added the limitations of self-report in the discussion.

The rate of missing disability data is also of concern because this data may not be missing at random in relation to disability status. I am also concerned about the difference in the sizes of the populations collected between genders - and am concerned that the healthy worker effect from the census methodology may be substantial. Are there other sources of data available to permit estimates of the proportion of males that may be missing? To help with transparency regarding missing data, Table S1 has been added to compare baseline characteristics of census respondents with study participants. The census is very thorough and participants do not need to be in their home at the time of the census survey to be counted. The disparity in numbers between men and women arises at study participation rather than at census inclusion. This is because they need to be seen at home to be asked (or a proxy to be asked) about disability status. We have added the number of female and male participants in Figure 1 to help clarify this.

The considerable missing data is rightly noted as a limitation, but may not be emphasized strongly enough or with sufficient detail about its potential effects on the analytic results. The rate of missing data for some areas is substantial and has a strong likelihood of affecting the results. I am glad to see that sensitivity analyses were performed and reported with the covariates. The observed relationship with hypertension and missing data underscores that this missing data may not be at random across the subgroups, and may bias the results. This limitation section may need to be stated more strongly given these observations.

Thank you for this comment. We have changed the wording of the discussion to emphasise the potential impact of the missing data on chronic disease on the findings.

Given the significant difference observed in the crude adult mortality rates between genders, it would be beneficial to age adjust one of the mortality rates to the other gender's age distribution to understand whether different age distributions within each gender may explain the observed difference. I note this type of sensitivity analysis was performed for men with and without disability, but not between genders. Given your later discussion of the survival paradox, it's to assess a potential role for different age distributions. We have now added a mortality rate ratio comparing the mortality of men and women controlling for age. This is in the fifth paragraph of the Results section.

For Table 4, the way the data is presented has the possibility to be misleading. Analyses of cause-specific mortality are an important area of analysis for this manuscript, especially as authors attempt to tie the paper's conclusions to it; the methodology for this table requires revision. The populations of people with no self-reported disability and self-reported disability are not of equal size, therefore, we would expect equal distribution within a cause of death category across these two groups. Because of this, simply comparing the percentages as done here is not informative. Instead, cause-specific adult mortality rates should be used for this comparison because this method will control for different underlying group sizes. A Risk Ratio or similar measure can be used to assess differences in cause-specific mortality rates between groups. Thank you for this observation. We have revised Table 4 to display the results as cause-specific mortality ratios.

The analyses in the paper don't directly support the conclusions. Disability does predict increased risk of mortality. However, the paper does not analyze why this association is present and does not directly identify healthcare access as a causal factor. I recognize that there are existing public health and access to healthcare challenges in this region, however, the analyses do not sufficiently link these challenges to the mortality outcomes, especially given the issues with the methodology used in Table 4. After revision of Table 4, authors should consider where there are differences in cause-specific mortality and improve the connection of the conclusions to these results. We have updated this paragraph so that we focus on the direct findings of our study.

There are areas throughout the paper where the language could be more specific and clear. For example, in lines 18-23 on page 10 where sentences begin with "This association..." and "This suggests...". It would be more clear to limit the use of "this" and specify the subject of the sentence. Thank you. We have updated the language and hope that it is now clearer.

This manuscript repeatedly states that the data is of high quality; I think is an overstatement of its quality. I understand that relative to other data sets from LIC countries it may be of higher quality. However, reliance on verbal autopsy and self-identification of disability does pose substantial risks to data quality. For example, certain causes of death are less likely to be identified through verbal autopsy, which likely biases this data. Additionally, missing information on 40% of the population is not trivial and as discussed in the limitation section, this missing data is likely not at random (due to difference in people not at home, etc.). We acknowledge that there are limitations to the data included in this paper as outlined in the discussion. However, as the study is set in a well-functioning DHS we do feel that we have high quality data on community-level mortality rates, which we comment on as a major strength of this study.

Many thanks again for your input, which has undoubtedly led to an improvement in the quality of this submission.

Dear Professor Walker,

Many thanks for your thoughtful comments on this submission. We have made a number of changes, including revisiting Figure 1 to make it clearer.

Please find below specific responses to your comments:

This is an interesting study from the Karonga HDSS in northern Malawi in which, in the annual censuses in 2014 and 2015, adults aged 18 and over were asked whether they experienced difficulty in any of the 6 functional domains of the Washington Group (WG) short set to collect disability statistics. Mortality data were then collected up until the 31 December 2017. The 16748 participants were followed up for a median of 29 months. 7.6% of the population surveyed reported disability and the overall mortality rate was 9.4/1000 person-years. People reporting disability had mortality rate 2.7 times higher than those without and 44.4% of people who died from NCDs had disability compared to 16.1% of those who died from HIV. While this is a large and impressive study, and these data are unique, there are some limitations, most of which the authors acknowledge.

Methods

The WG short set questions on disability have been asked in each census since 2014. Have the authors compared the answers from individuals in one year with the next year to check for consistency in responses and whether these don't change over time for those with a "fixed" disability, and do in others, who may have a progressive problem? Thank you for this suggestion. We have published data on the consistency of responses over the two consecutive years separately (see reference below). Interestingly, there was considerable movement of responses between "some difficulty" and "a lot of difficulty" over the course of the year, while "can't do at all" and "no difficulty" had higher consistency between the two years. Prynne JE, Dube A, Mwaiyeghele E *et al.* Self-reported disability in rural Malawi: prevalence, incidence, and relationship to chronic conditions [version 2; peer review: 1 approved with reservations, 1 not approved]. *Wellcome Open Res* 2019, 4:90

As the authors acknowledge excluding anyone who is away from home on the day of the census would mean that they were a lot more likely to miss men, particularly younger men, who may well have been away working.

The questions in the WG short set questions could be interpreted in a subjective way with inter-individual differences in interpretation of what constitutes “some difficulty” versus “a lot of difficulty” for example as the authors acknowledge. Was this area covered within training workshops for the field staff relating to how they would interpret the questions? In training workshops, the field team were advised to ask the Washington Group short set questions verbatim and with no prompting, so the responses reflected each participant’s own identification of whether they had “some difficulty” or “a lot of difficulty” in any particular domain.

Also, as the authors acknowledge the WG questions do not capture a comprehensive picture of disability, as compared to longer tools. People who were not picked up in round 1 (2014/2015) could be included if they were picked up in round 2 (2015/16) if they had turned 18 between the time of the 2 censuses or had moved into the area. Those recruited in round 2 would not have been followed up for as long.

The data on BMI, hypertension and diabetes were gathered in a survey between 2013 and 2015 and so are not contemporaneous, as acknowledged by the authors, with the disability survey. Also, as the authors acknowledge there are a lot of missing data for hypertension and diabetes as well as HIV status. This is a major limitation. In Table 1 no data on chronic disease were included for 4639 women and 3850 men as only participants with data on all three chronic diseases were included. As “number of chronic diseases” was a composite variable that required information on diabetes, hypertension, and HIV to calculate, there was considerable missing data. However, each of the individual diseases are also included separately in the table, which excluded only participants with missing data in that specific variable.

Results

I think that Figure 1 could be made a bit clearer. For the 39.6% who didn’t respond to questions on disability in round 1 were their demographic characteristics different from the 60.4% who did respond? For the 5893 not included at round 1 who provided disability data at round 2, what percentage is this of the total population who could have been recruited at round 2? Thank you for this observation. We have re-visited Figure 1 to clarify which participants at Round 2 were also seen at Round 1. We also recognised some minor errors in the calculations behind this figure which have now been rectified. 3860 participants who did not provide disability data in Round 1 went on to provide disability data in Round 2.

The fact that people could be recruited over 1 year apart would have greatly decreased the follow up time available for those recruited in round 2.

The 21.9% migration out of the HDSS is very high for a relatively short time period. This was an error and has been corrected – the correct figure is 9.5%.

Many thanks again for your time and thoughtful comments.

VERSION 2 – REVIEW

REVIEWER	Emily Lauer University of Massachusetts Medical School, USA
REVIEW RETURNED	07-Feb-2020

GENERAL COMMENTS	The revisions in the text surrounding the number of people who left the study due to migration do not agree with the numbers presented in the Figure. The text and figures should be reviewed again for accuracy and clarity. I do not think that the author revisions address the differential gender inclusion resulting from the study methods. Table S1 also shows that the men included in the study had a substantially
---

	different age profile than the men in the census and resulted in an older population of men included in the study. This has important implications for the male mortality rates, which does not appear to be fully addressed in the paper. My original comments about the description of the quality of the data still hold: This manuscript repeatedly states that the data is of high quality; I think is an overstatement of its quality. I understand that relative to other data sets from LIC countries it may be of higher quality. However, reliance on verbal autopsy and self-identification of disability does pose substantial risks to data quality. For example, certain causes of death are less likely to be identified through verbal autopsy, which likely biases this data. Additionally, missing information on 40% of the population is not trivial and as discussed in the limitation section, this missing data is likely not at random (due to difference in people not at home, etc.).
--	--

REVIEWER	Richard Walker Northumbria Healthcare NHS Foundation Trust, NE29 8NH, UK
REVIEW RETURNED	17-Feb-2020

GENERAL COMMENTS	The authors have addressed my major concerns and highlighted the major limitations. The reviewer provided a marked copy with additional comments. Please contact the publisher for full details.
--

VERSION 2 – AUTHOR RESPONSE

Reviewer: 1

Reviewer Name: Emily Lauer

Institution and Country: University of Massachusetts Medical School, USA

Please state any competing interests or state 'None declared': None

Please leave your comments for the authors below

1. The revisions in the text surrounding the number of people who left the study due to migration do not agree with the numbers presented in the Figure. The text and figures should be reviewed again for accuracy and clarity.

Response: many thanks for bringing this to our attention. There was a minor error in the figure which has now been corrected. The figure has also been amended for clarity.

2. I do not think that the author revisions address the differential gender inclusion resulting from the study methods. Table S1 also shows that the men included in the study had a substantially different age profile than the men in the census and resulted in an older population of men included in the study. This has important implications for the male mortality rates, which does not appear to be fully addressed in the paper.

Response: more information has been provided in the limitations section on the differential gender inclusion and what the likely impact would be on the results.

3. My original comments about the description of the quality of the data still hold:

- This manuscript repeatedly states that the data is of high quality; I think is an overstatement of its quality. I understand that relative to other data sets from LIC countries it may be of higher quality.

Response: We have taken out mention of high quality data.

However, reliance on verbal autopsy and self-identification of disability does pose substantial risks to data quality. For example, certain causes of death are less likely to be identified through verbal autopsy, which likely biases this data.

Response: We have mentioned as a limitation (in the discussion and in the summary section at the start) that verbal autopsy was used to determine cause of death, and this is liable to bias.

Additionally, missing information on 40% of the population is not trivial and as discussed in the limitation section, this missing data is likely not at random (due to difference in people not at home, etc.).

Response: We have stated in the limitations section that data were likely not be missing at random, which increased the chance of bias in results.

Reviewer: 2

Reviewer Name: Richard Walker

Institution and Country: Northumbria Healthcare NHS Foundation Trust, NE29 8NH, UK

Please state any competing interests or state 'None declared': None declared

Please leave your comments for the authors below

The authors have addressed my major concerns and highlighted the major limitations. Please see attached.

Were there follow-up meetings to clarify any issues about the Washington Group Questions?

Response: The team lead for the field interviewers had weekly meetings with the study lead who was able to give ongoing feedback and clarify any issues as they arose. The team lead would then ensure this information was disseminated throughout the field team.

There is a great deal of missing data for individual diseases (Hypertension, diabetes and HIV) as well as the composite measure of “chronic conditions”

Response: We have written in the limitations that there is missing data by individual disease and for the composite measure and that this may bias our results.